# Maskable Retentive Network for Video Moment Retrieval

Jingjing Hu
Hefei University of Technology,
School of Computer Science and
Information Engineering (School of
Artificial Intelligence)
xianhjj623@gmail.com

Dan Guo*
Hefei University of Technology,
Institute of Artificial Intelligence
(IAI), Hefei Comprehensive National
Science Center
guodan@hfut.edu.cn

Kun Li
CCAI,
Zhejiang University
kunli.hfut@gmail.com

Zhan Si
Anhui University, Department of
Chemistry and Centre for Atomic
Engineering of Advanced Materials
naa0528@stu.ahu.edu.cn

Xun Yang*
MoE Key Laboratory of
Brain-inspired Intelligent Perception
and Cognition, University of Science
and Technology of China
xyang21@ustc.edu.cn

Meng Wang*
Hefei University of Technology,
Institute of Artificial Intelligence
(IAI), Hefei Comprehensive National
Science Center
eric.mengwang@gmail.com

## ABSTRACT

Video Moment Retrieval (MR) tasks involve predicting the moment described by a given natural language or spoken language query in an untrimmed video. In this paper, we propose a novel Maskable Retentive Network (MRNet) to address two key challenges in MR tasks: cross-modal guidance and video sequence modeling. Our approach introduces a new retention mechanism into the multi-modal Transformer architecture, incorporating modality-specific attention modes. Specifically, we employ the Unlimited Attention for language-related attention regions to maximize cross-modal mutual guidance. Then, we introduce the Maskable Retention for video-only attention region to enhance video sequence modeling, that is, recognizing two crucial characteristics of video sequences: 1) bidirectional, decaying, and non-linear temporal associations between video clips, and 2) sparse associations of key information semantically related to the query. We propose a bidirectional decay retention mask to explicitly model temporal-distant context dependencies of video sequences, along with a learnable sparse retention mask to adaptively capture strong associations relevant to the target event. Extensive experiments conducted on five popular benchmarks ActivityNet Captions, TACoS, Charades-STA, ActivityNet Speech, and QVHighlights for MR tasks demonstrate the significant improvements achieved by our method over existing approaches. Code is available at https://github.com/xian-sh/MRNet.

## CCS CONCEPTS

• **Computing methodologies** → **Activity recognition and understanding**; • **Information systems** → **Multimedia and multimodal retrieval**.

*Corresponding authors

## KEYWORDS

Moment Retrieval, Maskable Retention, Transformer

**ACM Reference Format:**
Jingjing Hu, Dan Guo, Kun Li, Zhan Si, Xun Yang, and Meng Wang. 2024. Maskable Retentive Network for Video Moment Retrieval. In *Proceedings of the 32nd ACM International Conference on Multimedia (MM '24), October 28-November 1, 2024, Melbourne, VIC, Australia* ACM, New York, NY, USA, 10 pages. https://doi.org/10.1145/3664647.3680746

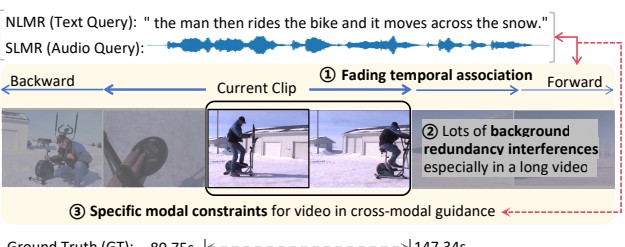

**Figure 1: An example of query-based video Moment Retrieval (MR). We address the *visual contextual association learning* challenge based on characteristics: ① Temporal association between video clips, ② Redundant background interference. And we address the *cross-modal guidance* challenge based on ③ Specific cross-modal attention constraints.**

## 1 INTRODUCTION

As an important application of multimedia and multimodal retrieval, query-guided video Moment Retrieval (MR) [8, 53] has attracted great attention from the research community in recent years [1, 5, 8, 12, 16–18, 22, 50, 57, 58, 63, 66, 67]. The MR task is initially proposed with the text as the query to retrieve the target moment within an untrimmed video, and this task setting still dominates the field of moment retrieval, which is called NLMR (*Natural Language Moment Retrieval*). As speech technology evolves, Xia *et al.* [53] explore the possibility of retrieving video with audio query and propose the ActivityNet Speech dataset and a new MR task, namely SLMR (*Spoken Language Moment Retrieval*). Fig. 1 shows their example,

given a sentence query "*the man then rides the bike and it moves across the snow*" or an audio query, the goal of MR tasks is to find the moment (80.75s-147.34s) that is semantically described by the query. Recently, some MR-related multi-task work such as moment retrieval and highlight detection (MR+HD) task has also been explored [14, 29], where HD refers to additionally determining the highlightness (or saliency) score of each clip in the predicted moments. Although these MR tasks are set up differently, they share some key challenges, *i.e.*, how to improve self-modal context modeling (especially "visual contextual association learning") and "cross-modal guidance" for better video content understanding and semantic alignment between language and video.

For the above two challenges, we rethink the MR tasks and highlight the following three characteristics: **(C1) Temporal association between video clips.** Existing works mostly emphasize some kind of relational modeling in terms of temporal sequence, *e.g.*, near-neighbor, recursive, local and global [9, 25, 30, 38]. However, as shown in Fig. 1, gradually decaying correlations are bidirectional along the timeline, these correlations are complex and non-linear [3], which may not be adequately expressed for example by weighting techniques like attention [47]. **(C2) Redundant background interference.** The background contains a lot of redundant information (Fig. 1) that can interfere with the recognition of the current event, and this redundancy is even worse in long videos, resulting in the key information semantically related to the target event being sparsely associated. Thus, additional de-redundancy strategies are important to highlight the key information in a video and improve the quality of moment retrieval. **(C3) Specific cross-modal attention constraints.** Since the video moment retrieval is a typical cross-modal task with the language query as guidance, and inspired by the success of Transformers [47] in cross-modal tasks [4, 6, 20, 35, 39, 70–73], many recent MR methods [14, 29, 62–64] adopt the Transformers to achieve the "cross-modal guidance" with no difference in the attention modes. However, we will no longer treat the different modalities indiscriminately, we impose the **C1 & C2** constraints on video-only attention region, but not on the other query-related regions, so as to improve visual modeling and maximize linguistic guidance. Concretely, in this work, we attempt to solve the "cross-modal guidance" challenge based on **C3** and the "video sequence modeling" challenge based on **C1 & C2**.

In this paper, we propose a new Maskable Retentive Network (MRNet) for MR tasks. The overview of MRNet is shown in Fig. 3. We first introduce a new retention mechanism [45] into the multimodal Transformer architecture. We divide the language and video attention map into four regions: $\mathcal{A}(q \rightarrow q)$, $\mathcal{A}(q \rightarrow v)$, $\mathcal{A}(v \rightarrow q)$ and $\mathcal{A}(v \rightarrow v)$, so that we can perform modality-specific operations based on the properties of the modality itself (**C3**): (1) As the linguistic semantics are highly condensed and in order to better maximize the guiding role of language, we set the Unlimited Attention mode for the language-related regions $\mathcal{A}(q \rightarrow q)$, $\mathcal{A}(q \rightarrow v)$ and $\mathcal{A}(v \rightarrow q)$, allowing queries can utilize visual details without restriction, and each video clip can response to each query, thus achieving cross-modal mutual guidance; (2) As for the pure video branch $\mathcal{A}(v \rightarrow v)$, we propose a new Maskable Retention mechanism to model two characteristics of video sequences (**C1 & C2**) together. Specifically, we firstly optimize the original retention mechanism [45] as a bidirectional decay mask along the timeline to

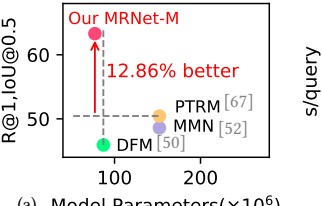
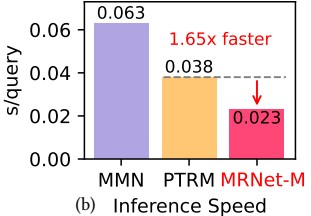

(a) Model Parameters(×10⁶)  (b) Inference Speed

**Figure 2: Comparison of recent best-performing multi-query trained MR methods in terms of model parameters and inference speed. Our MRNet-M demonstrates the optimal trade-off between model efficiency and accuracy.**

control the range within which the current video sequence token can be reviewed before and after with gradually fading attention, which can achieve the bidirectional recurrent sequence modeling effect. Second, in the video branch $\mathcal{A}(v \rightarrow v)$ of $T \times T$ size, a large number of redundant responses between the video clips serving as background may interfere with the highlighting of the target moment regions, and cause the key useful responses to be sparse, so we further propose a learnable sparse retention mask to adaptively highlight the strong responses and learn the key specific semantic associations related to the target event. In addition, to ensure the sparsity of the video attention, we design a self-supervised sparsity constraint loss to remove redundant responses.

To sum up, our proposed MRNet has parallel training capability like Transformers and aggregates bidirectional recursive reasoning capability like RNNs, enhanced visual modeling methodologies (like maskable retention, bidirectional decay, learnable sparse retention, nonlinearity, *etc.*). Moreover, our proposed architecture is applicable to NLMR, SLMR and MR+HD tasks and can be implemented in either single or multi-query training modes. In the validation of our experiments, the effect is maximized during multi-query training. The performance of our model is not only good, but also achieves a trade-off between model size and accuracy, as shown in Fig. 2.

The main contributions of this work can be summarized as follows: (1) We propose a new Maskable Retentive Network (MRNet) for three MR tasks (NLMR, SLMR and MR+HD) from a new perspective of deep video sequence modeling. It is a new attempt to introduce the retention mechanism into the multimodal Transformer framework for video sequence modeling, which is proven to have a significant performance advantage over other methods, and the architectural advantages such as scalability, recurrent sequential reasoning and sparsity. (2) We innovatively implement modality-specific attention modes, set Unlimited Attention for language-related attention regions to maximize cross-modal mutual guidance, and propose a new Maskable Retention for pure video branch for enhanced video sequence modeling. (3) We propose core video modeling methods: 1) bidirectional retention mask expresses temporal associations of video sequences (bidirectional, decaying, non-linear), 2) learnable sparse retention mask adaptively learns the strong associative responses related to the target event, and a new self-supervised loss is designed to constrain the sparsity of video attention region and remove redundant associations. (4) Extensive experiments are conducted on five benchmarks (ActivityNet

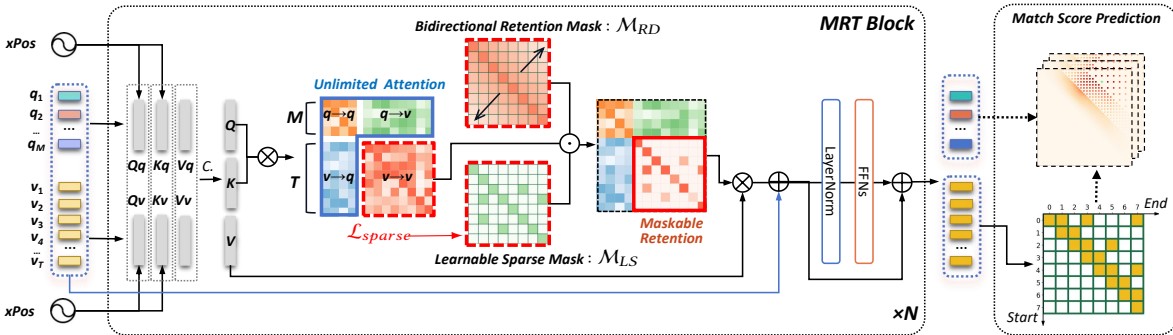

**Figure 3: Overview of our proposed Maskable Retentive Network (MRNet) architecture. The detailed description of *MRT block* is given in Sec. 3.2, after passing through multiple MRT blocks, the enhanced video features are used to create a 2D moment map in order to obtain the moment that best matches the query.**

Captions, TACoS, Charades-STA, ActivityNet Speech and QVHighlights datasets) for three MR tasks, and the excellent experimental results demonstrate the effectiveness of our proposed network.

## 2 RELATED WORK

**Video Moment Retrieval.** We review existing query-guided video Moment Retrieval (MR) methods in video sequence modeling. Generally, existing methods address the video modeling and query guidance with specific designs of cross-modal interaction [23, 24, 30, 37, 40, 44, 59, 62–64], feature fusion [16, 22, 54, 60, 61, 65–68], relational learning [9, 10, 23, 25, 31, 43, 52], *etc.*. And deep learning architecture technologies for video sequence modeling include clip-level sequential inference with RNNs [23, 25, 53, 56, 60, 62–64], moment relation modeling with CNNs [50, 52, 65–67] and GCNs [9, 23, 25, 43], and recently dominant semantic association learning between clip tokens with Transformers [14, 16, 19, 22, 25, 29, 30, 41, 44, 53, 62–64] in MR field. In this paper, we introduce a new retentive decay mechanism to model complex temporal correlation characteristics in video (*i.e.*, bidirectional, decaying, non-linear), and propose a new structural paradigm to address video moment retrieval tasks involving video and query interactions, which implements deep video retentive reasoning in a cross-modal environment to rethink the two key challenges of MR tasks as discussed in the Introduction.
**Retentive Network for Video Sequence Modeling.** Inspired by the success of the Transformers [47] in both natural language processing and computer vision [2, 6, 11, 34, 39, 48, 69], many effective transformer-based vision modeling strategies for MR tasks have been proposed [14, 16, 25, 29, 37, 49, 53, 63, 64]. Zhang *et al.* [64] adopt the visual language transformer encoder to jointly learn fine-grained associations between words and video clips, Xia *et al.* [53] propose the video-guided audio transformer decoding representation pre-training, Lei *et al.* [14] construct the Moment-DETR framework using transformer-based detection pipeline. The transformer-based moment retrieval structure enjoys efficient training and high performance. Recently, there have been some approaches such as RWKV [32] and RetNet [45] that have started to explore the advantages of combining Transformers and RNNs to improve the temporal inference capability of the model. In this paper, we attempt to introduce a new retention mechanism into transformer for

MR tasks. The new retention mechanism has the strengths for video sequence modeling such as parallel training like Transformers, bidirectional recurrent reasoning via the design of a new bidirectional decay mask, enhanced association learning through a learnable sparse retention mask and the nonlinearity regulation by updating the gate operation of retention.

## 3 OUR METHOD

### 3.1 Preliminary

*3.1.1 Problem Definition.* Given an untrimmed video consisting of a sequence of $T$ video clips, let its corresponding pre-extracted features be $\mathcal{V} = \{v_i\}_{i=1}^{T} \in \mathbb{R}^{T \times d^v}$, where $d^v$ is the clip feature dimension, and $Q = \{q_i\}_{i=1}^{M} \in \mathbb{R}^{M \times d^q}$ be a query set consisting of a number of $M$ language queries with the $d^q$-dim feature. The goal of both MR tasks NLMR (*natural language moment retrieval*) and SLMR (*spoken language moment retrieval*) is to predict the temporal boundaries ($\tau_{start}, \tau_{end}$) of target moment described by a given query $q$ (*text or audio modality*).

*3.1.2 Feature Encoding.* We linearly convert each query token and video clip token into a unified feature space. Thus, we obtain the query and video features $Q \in \mathbb{R}^{M \times d}$ and $\mathcal{V} \in \mathbb{R}^{T \times d}$, and represent their concatenated long input sequence as $Q||\mathcal{V}$, where $||$ denotes matrix concatenation operation. Considering the differences between video and language modalities, we independently add positional embedding for two modalities. Rotary position embedding method xPos [46] has decomposability and can better express the relative positional relationships of query and video sequences [42]. So we use it to encode two modalities:

$$
\begin{aligned}
Q_q &= (Q \cdot W_{Qq}) \odot \Theta_q, \ K_q = (Q \cdot W_{Kq}) \odot \overline{\Theta}_q \, ; \\
Q_v &= (\mathcal{V} \cdot W_{Qv}) \odot \Theta_v, \ K_v = (\mathcal{V} \cdot W_{Kv}) \odot \overline{\Theta}_v \, ; \\
V_q &= Q \cdot W_{Vq}, \quad V_v = \mathcal{V} \cdot W_{Vv} \, ; \\
Q &= Q_q||Q_v, \quad K = K_q||K_v, \quad V = V_q||V_v \, ,
\end{aligned}
\tag{1}
$$

where $\overline{\Theta}_{q/v}$ is the complex conjugate of $\Theta_{q/v} = e^{i\theta}$ [46] respectively, and $i$ is the index of sequence elements.

## 3.2 Maskable Retentive Network

In this section, we conduct modality-specific attention modes, that is, we set **Unlimited Attention** for language-related attention regions to maximize cross-modal mutual guidance, and perform a new **Maskable Retention** for video branch $\mathcal{A}(v \rightarrow v)$ for enhanced video sequence modeling. The core video modeling methods (Bidirectional Retention Mask, Learnable Sparse Retention Mask and self-supervised sparse constraint loss) in Maskable Retention will be described as follows.

*3.2.1 Video-Language Attention.* We follow the calculation of video-language attention $\mathcal{A} \in \mathbb{R}^{(M+T) \times (M+T)}$ in transformer. Eq. 2 divides the attention matrix into four blocks: $\mathcal{A}_{M \times M}, \mathcal{A}_{M \times T}, \mathcal{A}_{T \times M}$ and $\mathcal{A}_{T \times T}$, which represents four attention regions: $\mathcal{A}(q \rightarrow q)$, $\mathcal{A}(q \rightarrow v)$, $\mathcal{A}(v \rightarrow q)$ and $\mathcal{A}(v \rightarrow v)$. We adopt different attention modes for different attention regions based on the characteristics of the modality itself. That is, as videos often contain repetitive or closely similar visual images and sparsely response to the query, we apply the *Maskable Retention* in the visual branch $\mathcal{A}(v \rightarrow v)$ to remove redundant vision information and model retentive reasoning in video sequences and enhance temporal context association in the video. For query branch (other three attention regions), as the query semantics is relatively dense and doesn't reach a redundancy bound (each word in the queries has its own contextual linguistics), we leave them free to the query branch, so that queries and queries, and queries and video clip can be associated with each other without limitation in terms of retention calculation, thereby queries and video clips can provide comprehensive mutual guidance and semantic alignment as shown in Fig. 3.

$$
\begin{aligned}
\mathcal{A} = Q \cdot K^{\mathsf{T}} &= \begin{bmatrix} Q_q \\ Q_v \end{bmatrix} \cdot \begin{bmatrix} K_q^{\mathsf{T}} & K_v^{\mathsf{T}} \end{bmatrix} \\
&= \begin{bmatrix} Q_q \cdot K_q^{\mathsf{T}} & Q_q \cdot K_v^{\mathsf{T}} \\ Q_v \cdot K_q^{\mathsf{T}} & Q_v \cdot K_v^{\mathsf{T}} \end{bmatrix} = \begin{bmatrix} \mathcal{A}_{M \times M} & \mathcal{A}_{M \times T} \\ \mathcal{A}_{T \times M} & \mathcal{A}_{T \times T} \end{bmatrix}.
\end{aligned} \quad (2)
$$

The *Unlimited Attention* for query branch is tend to maximize the "linguistic guidance", and the *Maskable Retention* for visual branch $\mathcal{A}(v \rightarrow v)$ is designed to enhance the "vision modeling". Specifically, the additional optimization strategies *Maskable Retention* includes two important masking mechanisms, *i.e.*, the Bidirectional Retention Mechanism (expressed by a matrix $\mathcal{M}_{RD}$) and Learnable Sparse Retention Mask (expressed by a matrix $\mathcal{M}_{LS}$). The $\mathcal{M}_{RD}$ is designed for stronger temporal sequential reasoning and the $\mathcal{M}_{LS}$ is for highlighting key semantic associations in the video. Next we give the detailed explanations of them.

*3.2.2 Bidirectional Retention Mechanism.* For modeling the refined temporal context association between video sequences, we reconsider that a video is a sequence with retention relationships in time. Inspired by the powerful ability in sequence modeling of Retentive Network (RetNet) for language models [45], we introduce the retention mechanism into retentive reasoning in video sequence for better visual contextual association learning. Unlike the original RetNet that employing the unidirectional (only forward) retention, we incorporate a **bidirectional explicit retention decay masking** $\mathcal{M}_{RD}$ for $\mathcal{A}(v \rightarrow v)$, so that the current token can bidirectionally review temporal dependencies forward and backward:

$$
\text{BiRetention}(V_v) = (\mathcal{A}_{T \times T} \odot \mathcal{M}_{RD}) \cdot V_v ; \quad (3)
$$

$$
(\mathcal{M}_{RD})_{n,m} = \gamma^{|n-m|} , \mathcal{M}_{RD} \in \mathbb{R}^{T \times T}, \quad (4)
$$

where $\gamma$ is retention decay factor (*e.g.*, $\gamma = 0.98$), $n$ and $m$ are the row and column indices of matrix $\mathcal{M}_{RD}$, which combines bidirectional temporal masking and exponential decay along relative distance as shown in Fig. 3. Based on Eq. 4, we construct gradual decay effect from near to far on the video segment at the current time step and explicitly control the valid receptive field range of before and after the current token by setting a specific $\gamma$ value. This is used to perform recurrent retention reasoning in the video to enhance the model's temporally contextual learning ability like RNNs with transformer architecture.

*3.2.3 Learnable Sparse Retention Mask.* In addition to optimizing the temporal modeling capability of the model, we further optimize the model's ability to associate the semantics of video sequences. And we note the fact that in MR tasks, video sequences often contain a large amount of redundant background information, which can lead to excessive interference responses in the cross-modal attention map. And we argue that the strong responses in the attention map should be sparse, thus we further propose the Learnable Sparse Retention Mask to focus this issue for better "vision modeling".

To depict the key semantic associations between video sequence tokens, we innovatively introduce a **learnable sparse mask** $\mathcal{M}_{LS}$ that allows the model to focus more on relevant video tokens, learning an adaptive attention map to optimize for task-specific performance improvement. Then, we add a new self-supervised sparse constraint loss $\mathcal{L}_{sparse}$ for region $\mathcal{A}(v \rightarrow v)$ to reduce redundant noise interference within the video and learn more critical and useful information. We propose an auxiliary variable $W_{LS}$ of size $T \times T$ to build $\mathcal{M}_{LS}$, randomly initialize and optimize it during training. The calculation method for $\mathcal{M}_{LS}$ and sparsity constraint loss are:

$$
\mathcal{M}_{LS} = (1 - \text{I}_{T \times T}) \cdot \sigma(W_{LS}) + \text{I}_{T \times T} ; \quad (5)
$$

$$
\mathcal{L}_{sparse} = \frac{1}{T^2} \sum_{i=1}^{T} \sum_{j=1}^{T} \left| \sigma(W_{LS})_{i,j} \right|, \quad (6)
$$

here $\text{I}_{T \times T}$ is an identity matrix, we adopt the sigmoid function $\sigma$ with the threshold of 0.5 to scale $W_{LS}$ into $[0, 1]$, to build a soft mask. Here the mask $\mathcal{M}_{LS}$ is decomposed into two parts: self-token mask $\text{I}_{T \times T}$ and inter-token mask $(1 - I_{T \times T}) \cdot \sigma(W_{LS})$. The self-token mask is explicit and acts as a diagonal matrix. The minimization of the objective loss $\mathcal{L}_{sparse}$ of Eq. 6 is mainly imposed on the inter-token mask term. Finally, $\mathcal{L}_{sparse}$ serves as a regularizer to force the mask sparse, this drives some of the mask values towards 0, resulting in effective noise suppression. Acting $\mathcal{M}_{LS}$ on video branch $\mathcal{A}(v \rightarrow v)$, would force the model to learn a sparse attention map that retains only the key correspondences.

*3.2.4 Maskable Retentive Transformer Block.* We adopt the proposed maskable retention to enhance the video sequence inference ability of the model. With the learned matrices $\mathcal{M}_{RD}$ and $\mathcal{M}_{LS}$, we update the representation of language video attention $\mathcal{A}$:

$$
\mathcal{A}' = \begin{bmatrix} \mathcal{A}_{M \times M} & \mathcal{A}_{M \times T} \\ \mathcal{A}_{T \times M} & \underbrace{\mathcal{A}_{T \times T} \odot \mathcal{M}_{RD} \odot \mathcal{M}_{LS}}_{\text{Maskable Retention}} \end{bmatrix} ; \quad (7)
$$

$$
\text{MaskRetention}(X) = \text{Softmax}(\mathcal{A}') \cdot V . \quad (8)
$$

Table 1: The statistics of three widely used natural language moment retrieval (NLMR) datasets, the recent spoken language moment retrieval (SLMR) dataset, and the recent multi-task of moment retrieval and highlight detection (MR+HD) dataset.

| Datasets / Attributes | NLMR (the query is text in English) | | | SLMR (the query is audio in English) | MR+HD (multi-task) |
|---|---|---|---|---|---|
| | ANetCap [13] | TACoS [36] | Charades-STA [8] | ANetSpeech [53] | QVHighlights [14] |
| Domain | Open-world activity | Cooking | Indoors | Open-world activity | Vlog / News |
| Videos / Queries (Total) | 14,926 / 71,957 | 127 / 18,818 | 6,672 / 16,128 | 14,926 / 71,957 | 10,148 / 10,310 |
| Video length / Query length (Avg) | 117.61s / 15 words | 287.14s / 10 words | 30.59s / 7 words | 117.61s / 6.22s | 150s / 11 words |

We set up a $N$-layer MRT block to handle unified language and video sequence input $X$. A single-layer MRT block is defined as:

$$X' = X + \text{MaskRetention}(X);$$
$$\text{MRT}(X) = X' + \text{FFN}(\text{LayerNorm}(X')). \tag{9}$$

The enhanced language and video features are obtained after passing through $N$-layer MRT blocks, which are denoted as $X_N$. Then, the language and video are represented as $Q_N = \text{MRT}(X)[1:M;:] \in \mathbb{R}^{M \times d}$ and $\mathcal{V}_N = \text{MRT}(X)[M+1:M+T;:] \in \mathbb{R}^{T \times d}$, respectively.

## 3.3 Match Score Prediction

we follow the backend of 2D-TAN [66] to output the match score map of candidate moments and queries, with the query feature from the output of MRT block. Specifically, after passing through multiple MRT blocks, we obtaine enhanced language and video features, then we use the video features $\mathcal{V}_N \in \mathbb{R}^{T \times d}$ to build the 2D candidate moments map. We compute the Hadamard product of each query $q \in Q_N, Q_N \in \mathbb{R}^{M \times d}$, and 2D moments map to obtain the matching score map $S \in \mathbb{R}^{T \times T}$ with a total of $C$ moment candidates. To supervise the score map, we apply the binary cross-entropy loss to regress the IoU score of each moment:

$$\mathcal{L}_{iou} = \frac{1}{C} \sum_{i=1}^{C} (y_i \log(S_i) + (1 - y_i)\log(1 - S_i)). \tag{10}$$

We also adopt contrastive learning [52, 67] as an auxiliary positive and negative samples constraint:

$$\mathcal{L}_{contra} = -\left( \sum_{q \in Q^B} \log p(v_q|q) + \sum_{v \in \mathcal{V}^B} \log p(q_v|v) \right), \tag{11}$$

where $Q^B$ and $\mathcal{V}^B$ are the sets of queries and videos in a training batch. For $v_q \in \{v_q^+, v_q^-\}$, $v_q^+$ is the moment matched to query $q$ (solo positive sample), $v_q^-$ denotes the moment unmatched to $q$ in the training batch (multiple negative sample). $q_v \in \{q_v^+, q_v^-\}$, $q_v^+$ is the query matched to moment $v$, $q_v^-$ denotes the query unmatched to $v$ in the training batch. Thus the total training loss is

$$\mathcal{L} = \mathcal{L}_{iou} + \lambda_s \cdot \mathcal{L}_{sparse} + \lambda_c \cdot \mathcal{L}_{contra}, \tag{12}$$

where $\lambda_s$ and $\lambda_c$ are loss weights, and unsupervised loss term $\mathcal{L}_{sparse}$ we propose is defined in Eq. 6.

## 4 EXPERIMENTS

### 4.1 Datasets and Evaluation Metrics

*4.1.1 Datasets.* To evaluate the performance of our proposed MR-Net, we conduct experiments for three MR tasks (NLMR, SLMR, MR+HD) on five benchmarks as statistically described in Tab. 1. For NLMR task, three widely used datasets include: 1) **ActivityNet Captions (ANetCap)** [13] dataset contains $19,209$ videos from YouTube. Following the dataset partitioning [66], we use val_2 as the test set. Specifically, there are $37,417$ and $17,031$ sentence-moment pairs for train and test, respectively. 2) **TACoS** [36] dataset contains 127 videos where activities occur in the kitchen. It has 75 and 25 samples for training and testing, respectively. 3) **Charades-STA** [8] dataset contains $9,848$ videos that contains indoor activities. Following the same dataset split as [8] for fair comparisons, it has 12,408 and 3,720 samples for training and testing, respectively.

To validate the scalability of our model, we also conduct experiments on the recently proposed MR-related datasets. For SLMR task, there is currently only one public **ActivityNet Speech (ANet-Speech)** [53]. The queries of which are audios in English, obtained by volunteers reading text captions, and its dataset division is consistent with ANetCap dataset. For the MR+HD multi-task, The **QVHighlights** [14] dataset is recently proposed for joint moment retrieval and highlight detection tasks, it contains 10,148 open-world videos and 10,310 queries, we follow its data splits [14, 29].

*4.1.2 Evaluation Metrics.* Following the convention [8, 30, 66], we evaluate the performance of Moment Retrieval on two main metrics: 1) **Recall:** We adopt "R@$k$, IoU@$\mu$" as the evaluation metric, which represents the percentage of top-$k$ predicted moments whose tIoU (temporal Intersection Over Union) with the ground-truth moment is larger than $\mu$. We consider the following tIoU threshold values $\mu = \{0.1, 0.3, 0.5, 0.7\}$. 2) **mIoU:** We also adopt mIoU as the evaluation metric to count the average value of tIoU of all test samples. Additionally, for QVHighlights dataset, we follow official protocol [14]: mean average precision (**mAP**) is also a metric for Moment Retrieval (MR) and **HIT@1** indicates the hit ratio of the top-scoring clip to metric Highlight Detection (HD) task.

### 4.2 Implementation Details

*4.2.1 Feature Extraction.* For the preliminary extraction of video features, we use the C3D feature on ANetCap, ANetSpeech and TACoS from 2D-TAN [66], the I3D feature on Charades-STA from [30], as well as SlowFast/CLIP features on QVHighlights provided by [14] for a fair comparison. We set the number of sampled clips $T$ to 64 for ANetCap and Charades-STA, 128 for TACoS, and 75 for QVHighlights. For text features, following previous work [14, 52], we adopt the GloVe [33], BERT [39] feature and CLIP [35] feature (only on QVHighlights [14]). We extract the audio feature of ANetSpeech with the pre-trained Data2vec [2] model.

*4.2.2 Model Settings.* We implement two layers of MRNet block. For the back-end decoder setup, we adopt the same 2D proposal generation as 2D-TAN [66] and MMN [52] for the NLMR and SLMR tasks, and employ the same back-end (2-layer decoder and task loss settings) as Moment-DETR [14] and QD-DETR [29] for the joint

**Table 2: Evaluation results for ANetCap and TACoS datasets. The best results and second best performance are marked with bold and underline, respectively. We list the main visual processing backbones. "MRNet-S" and "MRNet-M" refer to the model with single-query training mode and multi-query mode, respectively.**

| Method | Backbones | Text | ANetCap R@1, IoU@ 0.3 | 0.5 | 0.7 | R@5, IoU@ 0.3 | 0.5 | 0.7 | mIoU | TACoS R@1, IoU@ 0.1 | 0.3 | 0.5 | R@5, IoU@ 0.1 | 0.3 | 0.5 | mIoU |
|---|---|---|---|---|---|---|---|---|---|---|---|---|---|---|---|---|
| VSLNet [63] | Transformer | GloVe | 63.16 | 43.22 | 26.16 | - | - | - | 43.19 | - | 29.61 | 24.27 | - | - | - | 24.11 |
| 2D-TAN [66] | CNN | GloVe | 59.45 | 44.51 | 26.54 | 85.53 | 77.13 | 61.96 | - | 47.59 | 37.29 | 25.32 | 70.31 | 57.81 | 45.04 | - |
| CSMGAN [25] | RNN/GCN | GloVe | 49.11 | 29.15 | 77.43 | 59.63 | - | 33.90 | 27.09 | 53.98 | 41.22 | - | | | | |
| CPNet [16] | Transformer | GloVe | - | 40.56 | 21.63 | - | - | - | 40.65 | - | 42.61 | 28.29 | - | - | - | 28.69 |
| VSLNet-L [62] | Transformer | GloVe | 62.35 | 43.86 | 27.51 | - | - | - | 44.06 | - | 47.11 | 36.34 | - | - | - | 36.61 |
| MS-2D-TAN [65] | CNN | GloVe | 61.04 | 46.16 | 29.21 | 87.30 | 78.80 | 60.85 | - | 49.24 | 41.74 | 34.29 | 78.33 | 67.01 | 56.76 | - |
| MSAT [64] | Transformer | - | - | 48.02 | 31.78 | - | 78.02 | 63.18 | - | - | 48.79 | 37.57 | - | 67.63 | 57.91 | - |
| RaNet [9] | GCN/CNN | GloVe | - | 45.59 | 28.67 | - | 75.93 | 62.97 | - | - | 43.34 | 33.54 | - | 67.33 | 55.09 | - |
| FVMR [10] | Transformer | GloVe | 60.63 | 45.00 | 26.85 | 86.11 | 77.42 | 61.04 | - | 53.12 | 41.48 | 29.12 | 78.12 | 64.53 | 50.00 | - |
| MMN [52] | CNN | BERT | 65.05 | 48.59 | 29.26 | 87.25 | 79.50 | 64.76 | - | 51.39 | 39.24 | 26.17 | 78.03 | 62.03 | 47.39 | - |
| MGPN [44] | CNN/Transformer | GloVe | - | 47.92 | 30.47 | - | 78.15 | 63.56 | - | - | 48.81 | 36.74 | - | 71.46 | 59.24 | - |
| DCLN [61] | CNN | GloVe | 65.58 | 44.41 | 24.80 | 84.65 | 74.04 | 56.67 | - | 65.16 | 44.96 | 28.72 | 82.40 | 66.13 | 51.91 | - |
| SPL [23] | Transformer/GCN | GloVe | - | 52.89 | 32.04 | - | 82.65 | 67.21 | - | - | 42.73 | 32.58 | - | 64.30 | 50.17 | - |
| VGCL [53] | Transformer/RNN | GloVe | 60.57 | 42.96 | 25.68 | - | - | - | 43.34 | - | - | - | - | - | - | - |
| MA3SRN [22] | Transformer/GCN | GloVe | - | 51.97 | 31.39 | - | 84.05 | 68.11 | - | - | 47.88 | 37.65 | - | 66.02 | 54.27 | - |
| PTRM [67] | CNN | BERT | 66.41 | 50.44 | 31.18 | - | - | - | 47.68 | - | - | - | - | - | - | - |
| CRaNet [43] | GCN/CNN | GloVe | - | 47.27 | 30.34 | - | 78.84 | 63.51 | - | - | 47.86 | 37.02 | - | 70.78 | 58.39 | - |
| BMRN [40] | Transformer/CNN | BERT | - | 48.47 | 31.15 | - | 81.37 | 64.44 | - | - | - | - | - | - | - | - |
| PLN [68] | CNN | GloVe | 59.65 | 45.66 | 29.28 | 89.66 | 76.65 | 63.06 | 44.12 | 53.74 | 43.89 | 31.12 | 75.56 | 65.11 | 52.89 | 29.70 |
| DFM [50] | CNN | GloVe | 58.84 | 45.92 | 32.18 | - | - | - | - | - | 40.04 | 28.57 | - | - | - | 27.35 |
| MRNet-S (Ours) | Retentive Network | GloVe | 71.92 | 54.91 | 31.58 | 91.28 | 85.98 | 73.73 | 51.48 | 70.01 | 53.74 | 37.54 | 88.05 | 77.76 | 64.76 | 37.41 |
| MRNet-S (Ours) | Retentive Network | BERT | 73.40 | 55.40 | 32.46 | 90.82 | 85.24 | 72.23 | 51.80 | 69.43 | 54.49 | 37.84 | 87.60 | 77.26 | 64.73 | 37.51 |
| MRNet-M (Ours) | Retentive Network | GloVe | 74.74 | 59.40 | 39.12 | 90.91 | 85.12 | 73.45 | 54.86 | 71.71 | 55.41 | 38.54 | 89.43 | 77.18 | 64.78 | 38.19 |
| MRNet-M (Ours) | Retentive Network | BERT | **77.57** | **63.30** | **42.68** | **91.60** | **86.25** | **75.92** | **57.62** | **71.98** | **56.16** | **41.31** | 89.33 | **78.43** | **66.21** | **39.45** |

**Table 3: Evaluation results for Charades-STA dataset.**

| Method | R@1, IoU@ 0.3 | 0.5 | 0.7 | R@5, IoU@ 0.3 | 0.5 | @0.7 | mIoU |
|---|---|---|---|---|---|---|---|
| DRN [60] | - | 53.09 | 31.75 | - | 89.06 | 60.05 | - |
| TMLGA [38] | 67.53 | 52.02 | 33.74 | - | - | - | - |
| LGI [30] | 72.96 | 59.46 | 35.48 | - | - | - | 51.38 |
| BPNet [54] | 65.48 | 50.75 | 31.64 | - | - | - | 46.34 |
| CPNet [16] | - | 60.27 | **38.74** | - | - | - | 52.00 |
| MS-2D-TAN [65] | - | 60.08 | 37.39 | - | 89.06 | 59.17 | - |
| FVMR [10] | - | 55.01 | 33.74 | - | **89.17** | 57.24 | - |
| $I^2N$ [31] | - | 56.61 | 34.14 | - | 81.48 | 55.19 | - |
| PLN [68] | 68.60 | 56.02 | 35.16 | 94.54 | 87.63 | 62.34 | 49.09 |
| $M^2DCapsN$ [26] | - | 55.03 | 34.61 | - | 84.33 | 63.71 | - |
| MRNet-S (Ours) | 74.17 | 60.03 | 38.25 | 96.29 | 88.79 | **69.92** | 53.01 |
| MRNet-M (Ours) | **74.65** | **60.30** | 38.20 | 96.51 | **89.09** | 69.52 | **53.27** |

**Table 4: Evaluation results for ANetSpeech dataset. † denotes the result reproduced by us.**

| Method | R@1, IoU@ 0.3 | 0.5 | 0.7 | R@5, IoU@ 0.3 | 0.5 | 0.7 | mIoU |
|---|---|---|---|---|---|---|---|
| VSLNet | 46.75 | 29.08 | 16.24 | - | - | - | 34.01 |
| VGCL [53] | 49.80 | 30.05 | 16.63 | - | - | - | 35.36 |
| SIL [51] | 49.46 | 30.26 | 15.22 | 82.28 | 63.73 | 35.48 | 34.52 |
| VSLNet† | 51.02 | 30.38 | 17.45 | - | - | - | 37.04 |
| MMN† | 51.98 | 35.69 | 20.77 | 85.46 | 75.29 | 56.87 | 37.81 |
| MRNet-S (Ours) | 66.72 | 49.80 | 29.12 | 90.24 | 84.05 | 72.53 | 48.12 |
| MRNet-M (Ours) | **70.41** | **54.61** | **34.00** | **90.82** | **85.19** | **73.10** | **51.31** |

**Table 5: Evaluation results for QVHighlights dataset. As one query per video, we only test the single-query training mode.**

| Method | MR R@1, IoU@ @0.5 | @0.7 | mAP @0.5 | @0.75 | Avg. | HD >= Very Good mAP | HIT@1 |
|---|---|---|---|---|---|---|---|
| MCN [1] | 11.41 | 2.72 | 24.94 | 8.22 | 10.67 | - | - |
| CAL [7] | 25.49 | 11.54 | 23.40 | 7.65 | 9.89 | - | - |
| CLIP [35] | 16.88 | 5.19 | 18.11 | 7.00 | 7.67 | 31.30 | 61.04 |
| XML [15] | 41.83 | 30.35 | 44.63 | 31.73 | 32.14 | 34.49 | 55.25 |
| XML+ [14] | 46.69 | 33.46 | 47.89 | 34.67 | 34.90 | 35.38 | 55.06 |
| Moment-DETR [14] | 52.89 | 33.02 | 54.82 | 29.40 | 30.73 | 35.69 | 55.60 |
| UMT [27] | 56.23 | 41.18 | 53.83 | 37.01 | 36.12 | 38.18 | 59.99 |
| MH-DETR [55] | 60.05 | 42.48 | 60.75 | 38.13 | 38.38 | 38.22 | 60.51 |
| QD-DETR [29] | 62.40 | 44.98 | 62.52 | 39.88 | 39.86 | 38.94 | 62.40 |
| UniVTG [21] | 58.86 | 40.86 | 57.60 | 35.59 | 35.47 | 38.20 | 60.96 |
| MRNet-S (Ours) | **64.85** | **46.63** | **65.11** | **42.06** | **41.63** | **39.69** | **63.55** |

MR+HD task. For training, we control the sparse ratio of $\mathcal{M}_{LS}$ to 25%, and set the loss weight $\lambda_s$ of $\mathcal{L}_{sparse}$ to 10.0 for all datasets. The contrastive loss weight $\lambda_c$ is consistent with MMN [52], *i.e.*, 0.1 for ANetCap and TACoS, 0.05 for Charades-STA. We use AdamW [28] optimizer with a learning rate $8 \times 10^{-4}$ for ANetCap, $1 \times 10^{-3}$ for Charades-STA and $15 \times 10^{-4}$ for TACoS, and batch size 12 for all datasets. The setup for ANetSpeech is the same as for ANetCap.

*4.2.3 Training Modes.* In MR task, one-to-one (1 query to 1 video) training is known as single- query training [63, 66, 68], many-to-one (many queries to one video) training is known as multi-query training [24, 50, 52, 67]. We test the effect of single-query training and adopt a multi-query training model to maximize the linguistic guidance from queries. Noting that during inference, regardless of any training mode, the evaluation is a fair single query input

to predict a uniquely corresponding moment, consistent with the convention of the MR tasks [8, 16, 66].

## 4.3 Comparisons to the State-of-The-Art

*4.3.1 ANetCap and TACoS for NLMR task.* These two datasets are the ones most evaluated in MR tasks. The ANetCap dataset is the largest open-domain dataset for MR task, the video duration of TACoS dataset is the longest. We compare our MRNet with recent NLMR state-of-the-art (SOTA) methods [9, 10, 16, 22, 23, 25, 40, 43, 44, 50, 52, 53, 61–68]. Tab. 2 demonstrates the absolute advantages of our method compared to other methods across the ANetCap and TACoS datasets. Our MRNet-S has reached SOTA performance, when we adopt a multi-query training mode, our model's performance has been further improved, language guidance is maximized. Specifically, our MRNet-M achieves a new SOTA performance on all metrics, such as R@1, IoU@0.5 (63.30 on ANetCap, 41.31 on TAcoS), and mIoU (57.62 on ANetCap, 39.45 on TAcoS).

*4.3.2 Charades-STA for NLMR task.* The Charades-STA dataset has the shortest query descriptions (avg. 7.22 words), and shortest video durations (avg. 30.6s), it requires the identification of more subtle human movements and the extraction of more linguistic information. Despite this limitation, our MRNet achieves comparable performance to other state-of-the-art methods on the Charades-STA dataset, as indicated in Tab. 3. Specifically, our MRNet-M outperforms other methods in terms of R@1, IoU0.3, R@1, IoU@0.5, and mIoU metrics with values of 74.65, 60.30, and 53.27, respectively.

*4.3.3 ANetSpeech for SLMR task.* We validate the performance advantages of our MRNet for SLMR task on the ANetSpeech dataset [53], when the queries are audios in English. We reproduce the evaluations of the typical MR methods VSLNet [63] and MMN [52], and compare our model with other state-of-the-art methods VGCL [53] and SIL [51]. The results are shown in Tab. 4, our MRNet-M currently has optimal performances on ANetSpeech dataset, *e.g.*, 34.00 on R@1, IoU@0.7, and 51.31 on mIoU, which has significant performance improvement compared to other MR methods, demonstrating our model's good scalability on MR tasks.

*4.3.4 QVHighlights for the MR+HD multi-task.* We further validate our model on the multi-task MR-related dataset QVHighlights (MR+HD) [14]. This joint task includes both finding relevant moments and predicting the highlighted score of each clip in them. The comparisons with existing works are detailed in Tab. 5. From the results, our model demonstrates performance on par with SOTA models, achieving an $R@1, IoU@0.7$ of 46.63 for moment retrieval, and a HIT@1 of 63.55 for highlight detection. It further affirms the robust universality of our model on MR+HD tasks.

## 4.4 Ablation Studies

In this part, we evaluate the impact of various factors of the proposed MRNet to answer the following research questions (all experiments are based on MRNet-M): **Q1:** How do two core mask components $\mathcal{M}_{RD}$ and $\mathcal{M}_{LS}$ contribute to the performance of our MRNet? **Q2:** Is a bidirectional retention more effective than a unidirectional one? Which non-linear operation is more effective: SoftMax or Swish Gate? **Q3:** Will the self-supervised sparsity loss be beneficial? How to set this loss's weight $\lambda_s$ and the sparsity ratio $\alpha$

of the $\mathcal{M}_{LS}$ to be optimal? **Q4:** Which masking mode is more effective: only masking video or full masking on the entire cross-modal map? An qualitative case of cross-modal retention map. **Q5:** The quantitative case analysis of video moment retrieval.

*4.4.1 Main Module Analysis (Q1).* We verify the contributions of the main two mask components $\mathcal{M}_{RD}$ and $\mathcal{M}_{LS}$ of our proposed MRNet framework, *i.e.*, bidirectional retention mask module and learnable sparse retention mask module. As shown in Tab. 6, we combine the retentive network backbone and the 2D proposal backend of 2D-TAN [66] as the baseline (Row 1), namely removing $\mathcal{M}_{RD}$ and $\mathcal{M}_{LS}$ from the model. In our work, the proposed baseline is a new architecture, it is equipped with RetNet [45] in a cross-modal Transformer architecture, which integrates the strengths such as Transformer (Parallel learning, Eq. 2), RetNet (Rerecursive inference, Eq. 3), nonlinearity regulation (Eq. 8). Compared with the SOTA methods in Tab. 2, the performance of the baseline is good enough. Furthermore, with the independent addition of the bidirectional retention mask $\mathcal{M}_{RD}$ or the learnable sparse retention mask $\mathcal{M}_{LS}$ on the baseline, the model achieves a considerable performance improvement. When we model bidirectional video temporal associations and then add sparsity mask as a constraint to erase unnecessary dependencies in video sequences, the performance of the model is further improved. The results show that both mask modules of MRNet are coordinated and compatible, and contribute to achieving significant improvement on the representation learning and generalization performance of the MR model.

*4.4.2 Retention Mechanism Analysis (Q2).* There are two operations that we optimize in the Retention module as shown in Fig. 5: (1) From unidirectional to bidirectional. If we use unidirectional retention mask instead of bidirectional $\mathcal{M}_{RD}$ (the unidirectional $\mathcal{M}_{RD}$ has only forward mask as $(\mathcal{M}_{RD})_{n,m} = \gamma^{n-m} \ if \ n \geq m; \ else \ 0.$), the performance of the model is compromised, *e.g.*, R@1, IoU@0.5 decreased by 1.60 point, this indicates that bidirectional $\mathcal{M}_{RD}$ is indeed stronger in its ability to model video sequences than unidirectional in our MRNet. (2) SoftMax over Gating: we define the "MaskRetention" approach in Eq. 8 that adopts the SoftMax operation (Transformer-like), while Swish Gate is specifically designed to handle the model's nonlinearity in original Retention [45], we use SoftMax instead, the Fig. 5 (right) shows the advantages of the SoftMax operation over the Swish Gate operation, which promises both complexity reduction and performance improvement.

*4.4.3 Masking Sparsity Constraint (Q3).* We evaluate the effect of self-supervised loss $\mathcal{L}_{sparse}$ that is used to constrain the sparsity of learnable mask $\mathcal{M}_{LS}$. From Tab. 7, $\mathcal{L}_{sparse}$ is effective in improving the performance of the model. We further give the ablation of two important hyperparameters of $\mathcal{M}_{LS}$ in Fig. 6, *i.e.*, the loss weight $\lambda_s$ of $\mathcal{L}_{sparse}$ and the sparsity ratio $\alpha$ (%) of $\mathcal{M}_{LS}$, where $\alpha$ controls the mask's proportion of $\mathcal{M}_{LS}$ during training by setting the average convergence cutoff threshold for $\mathcal{M}_{LS}$ at $0.5\alpha$. The optimal setting of $\lambda_s$ is 10.0, and the sparsity ratio $\alpha$ is 25%.

*4.4.4 Masking Mode Analysis (Q4).* We discuss different masking modes for the vision-language attention map. The "No masking (vanilla attention)", "Only masking video" and "Full masking" modes in Fig. 7 correspond to the first, second and last rows of Tab. 8, respectively. In our solution, we leave the query free and focus

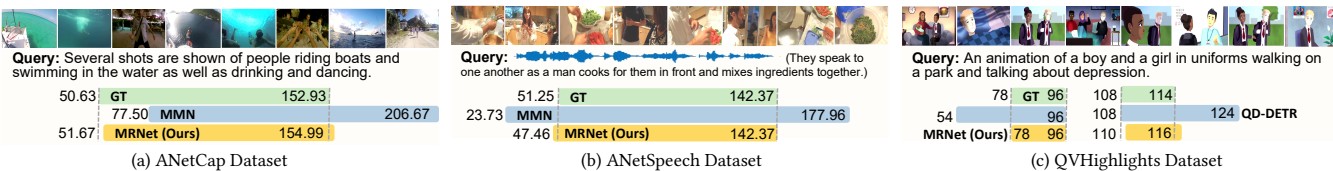

(a) ANetCap Dataset  (b) ANetSpeech Dataset  (c) QVHighlights Dataset

**Figure 4: Qualitative results. Compared with MMN and QD-DETR, our MRNet can locate more accurate temporal regions.**

**Table 6: R@1 results of MRNet w/o mask $\mathcal{M}_{RD}$ and $\mathcal{M}_{LS}$.**

| $\mathcal{M}_{RD}$ | $\mathcal{M}_{LS}$ | ANetCap | | | Charades-STA | | |
|---|---|---|---|---|---|---|---|
| | | 0.5 | 0.7 | mIoU | 0.5 | 0.7 | mIoU |
| ✗ | ✗ | 58.59 | 38.93 | 54.72 | 57.80 | 36.72 | 51.25 |
| ✓ | ✗ | 61.06 | 40.28 | 56.04 | 58.95 | 37.31 | 52.57 |
| ✗ | ✓ | 61.33 | 40.39 | 56.23 | 59.03 | 37.28 | 52.54 |
| ✓ | ✓ | **63.30** | **42.68** | **57.62** | **60.27** | **38.39** | **53.14** |

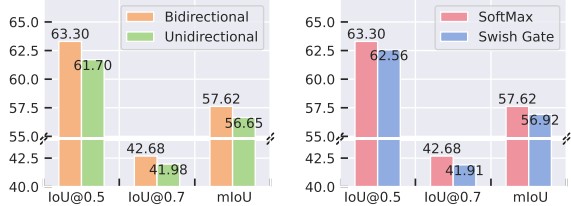

**Figure 5: Ablation studies on Directionality of $\mathcal{M}_{RD}$, and Nonlinear operation of MaskRetention on ANetCap dataset.**

**Table 7: R@1 results of MRNet w/o sparsity loss $\mathcal{L}_{sparse}$.**

| $\mathcal{M}_{LS}$ | $\mathcal{L}_{sparse}$ | ANetCap | | | Charades-STA | | |
|---|---|---|---|---|---|---|---|
| | | 0.5 | 0.7 | mIoU | 0.5 | 0.7 | mIoU |
| ✗ | ✗ | 61.06 | 40.28 | 56.04 | 58.95 | 37.31 | 52.57 |
| ✓ | ✗ | 62.79 | 41.86 | 57.04 | 60.00 | 38.23 | 52.98 |
| ✓ | ✓ | **63.30** | **42.68** | **57.62** | **60.27** | **38.39** | **53.14** |

on the sparsity of video, with the core idea of removing redundancy within the video and making the most of query semantics. From Tab. 8, we test different mask combinations, whenever it comes to query-related masking, there is some performance degradation, which confirms our above core idea, *i.e.*, only "masking "$\mathcal{A}(v \rightarrow v)$ is optimal. Moreover, from Fig. 7, if we mask the whole vision-language map, the performance is higher than that with "No masking", this indicates that vanilla attention brings redundant correlations ($T \times T$) and hurts performance. It is reasonable and effective to consider removing redundancy between video attention responses. The visualized mask retention map of our solution "Only masking video" shows that the retention mask is well concentrated in the groundtruth (GT) region, the response in the other regions is close to 0, realizing effective noise suppression.

*4.4.5 Qualitative Results (Q5).* In order to better understand the moment retrieval results of our MRNet, we take three videos from ANetCap, ANetSpeech, and QVHighlights datasets as examples and display the qualitative results of them in Fig. 4. Compared with the strong MR methods MMN [52] and QD-DETR [29], our proposed MRNet has almost no gap compared to GroundTruth (GT)

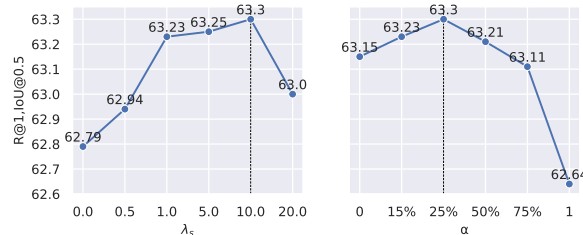

**Figure 6: Ablation studies on the sparsity loss weight $\lambda_s$ of $\mathcal{L}_{sparse}$ and the sparsity ratio $\alpha$ of $\mathcal{M}_{LS}$ on ANetCap dataset.**

**Table 8: R@1 results for masking different attention region on ANetCap dataset. The first row refers to vanilla attention.**

| $\mathcal{A}(v \rightarrow v)$ | $\mathcal{A}(q \rightarrow q)$ | $\mathcal{A}(v \rightarrow q)$ | $\mathcal{A}(q \rightarrow v)$ | IoU0.3 | IoU0.5 | IoU0.7 |
|---|---|---|---|---|---|---|
| ✗ | ✗ | ✗ | ✗ | 73.39 | 58.59 | 38.93 |
| ✓ | ✗ | ✗ | ✗ | **77.57** | **63.30** | **42.68** |
| ✓ | ✓ | ✗ | ✗ | 76.73 | 62.18 | 41.69 |
| ✓ | ✓ | ✓ | ✗ | 75.56 | 61.47 | 41.02 |
| ✓ | ✓ | ✓ | ✓ | 75.09 | 60.40 | 40.39 |

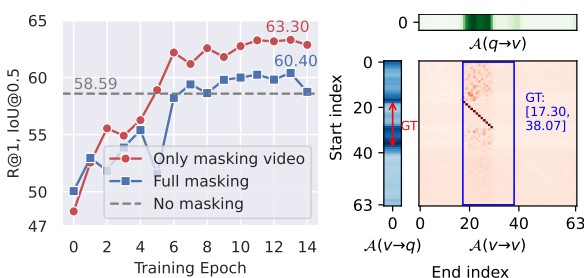

**Figure 7: Masking modes analysis on ANetCap dataset. Convergence curves and visualization of "Only masking video".**

across three datasets, which indicates that broad adaptability of our approach to MR-related tasks. Overall, the visualization results demonstrate the effectiveness of our proposed MRNet.

## 5 CONCLUSION

In this paper, we propose a compact single-stream Maskable Retentive Network (MRNet) as a new structural paradigm for addressing video Moment Retrieval tasks involving video and query interactions, which implements video retentive reasoning in a cross-modal environment. The effectiveness of MRNet framework has been demonstrated on five standard benchmarks for three MR tasks.

# ACKNOWLEDGMENTS

This work was supported by the National Natural Science Foundation of China (62272144,U22A2094,72188101,62020106007 and U20A20183), the Major Project of Anhui Province (202203a05020011), and the Fundamental Research Funds for the Central Universities (JZ2024HGTG0309, JZ2024AHST0337 and JZ2023YQTD0072).

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
