# OpenReview forum: "Maskable Retentive Network for Video Moment Retrieval"
_acmmm.org/ACMMM/2024/Conference — MM2024 Poster_

### Official Review · Reviewer_V685 · 2024-05-22

**Rating:** 3
**Confidence:** 3

**Summary:**

In this paper, a novel Maskable Retentive Network (MRNet) is proposed to address two key difficulties in Video Moment Retrieval tasks: cross-modal guidance and video sequence modeling. A novel retention mechanism is introduced into classical multimodal Transformer and divide the attention into four areas. The Unlimited Attention is leveraged to language-related attention regions to maximize cross-modal mutual supervision and the Maskable Retention is employed for video-only attention region to boost video sequence modeling. In addition, extensive experiments results verify the important improvement of MRNet compared with existing methods.

**Strengths:**

In this paper, two learnable masks proposed for the only-video attention region can enhance the model's ability to learn the temporal context of the video, as well as the redundant background information of the video is removed, so that the strong responses on the attention map become sparse, and then highlight the relevant video tokens. The comparison experiment is fully conducted, and the ablation study is also relatively comprehensive, which can clearly highlight the impact of some important module in the MRNet.

**Limitations:**

1.	This paper has not clearly expressed the benefits of dividing attention map into four regions. If only to perform modality-specific modeling with the properties of the modality itself, why we not directly utilize the vanilla cross/self attention mechanisms.
2.	This paper seems to only concentrate on the only-video attention ($A(v \rightarrow v)$) region and does not make some modification to query-video ($A(q \rightarrow v)$) and video-query ($A(v \rightarrow q)$) attention regions, but the latter two is also very essential for cross-modal feature fusion.
3.	In formula 1, only Q and V are added with position coding. Why is the calculation of V (V_q and V_v) not appended with position coding?
4.	Formula 11 doesn’t seem to be a representation of the contrastive learning loss function, since there is no obvious contrast between positive and negative samples, it is more like cross entropy loss.
5.	There are some small problems such as formatting in the article, as described below:
   A.	In line 95, the “highlightness” dose not seems to be a valid derivative.
   B.	In line 454, the abbreviation “MRT” have already been explained before (line 287) and need not be explained more than once.
   C.	In line 487, the citations [45, 46] are actually the same paper.

**Suitability:**

3

---

### Official Review · Reviewer_VMLT · 2024-05-22

**Rating:** 3
**Confidence:** 4

**Summary:**

This paper introduces a novel deep learning architecture, Maskable Retentive Network (MRNet), designed to address the challenges in Video Moment Retrieval (MR) tasks. MR tasks involve locating specific moments within untrimmed videos based on natural language or spoken language queries. The authors highlight two key challenges in MR: cross-modal guidance and video sequence modeling.
To tackle these challenges, MRNet integrates a new retention mechanism into the multimodal Transformer architecture. This mechanism employs modality-specific attention modes, including Unlimited Attention for language-related regions to enhance cross-modal guidance and a Maskable Retention mechanism for the video-only attention region to improve video sequence modeling. The proposed system also introduces a bidirectional decay retention mask and a learnable sparse retention mask to model temporal-distant context dependencies and capture strong associations relevant to the target event, respectively.
The authors conducted extensive experiments on five benchmarks: ActivityNet Captions, TACoS, Charades-STA, ActivityNet Speech, and QVHighlights. The results demonstrate that MRNet significantly outperforms existing approaches, showcasing its effectiveness in MR tasks.

**Strengths:**

1. The paper introduces the retention mechanism, including modality-specific attention modes like Unlimited Attention and Maskable Retention, which is interesting.
2. The paper provides extensive experimental results across multiple benchmarks, which validate the effectiveness of the proposed MRNet.

**Limitations:**

The readability of this paper is reduced due to the following issues.
1. The paper is full of long and difficult sentences, which are difficult to understand and read. For example, lines 145-150, lines 365-372.
2. Why are the characteristics of video sequences bidirectional, attenuated, and nonlinear? The paper does not give a very clear explanation.
3. The symbols of the formula are not clearly defined. For example, X in Formula 3 is not defined. The (·) symbol in formula (1) is not explained clearly.
4. There are many typo errors in the article, such as two "learnables" on lines 382-383 and two R3.3 on lines 660.
5. Why is the attention score between text and video not masked? In fact, the text also contains tokens that are more relevant to the video and tokens that are less relevant, which the author does not seem to have considered.
6. How is the matching score map in line 478 obtained, and how is the query feature obtained?
7. The code is not provided, and I doubt the reproducibility of the article’s results.
8. While the paper shows excellent performance on specific benchmarks, it is unclear of robustness of the model. It is recommended to test on ood-datasets like Charades-CD [1] or Charades-CG [2] to test the generalization ability of the model.
9. It’s weird to me. From the beginning, the input for text is the sequence of sentence feature and the input for video is the sequence of video clips. These two types of features are not at the same granularity to attend to each other.
10. The CPL in Table 3 is weakly supervised, but your method is fully supervised. It is unreasonable to compare them together.
11. What is the difference between single-query training mode and multi-query mode?

[1] A Closer Look at Temporal Sentence Grounding in Videos

[2] Compositional Temporal Grounding with Structured Variational Cross-Graph Correspondence Learning

**Suitability:**

3

---

### Official Review · Reviewer_WfJx · 2024-05-23

**Rating:** 4
**Confidence:** 3

**Summary:**

This paper introduces the Maskable Retentive Network (MRNet), a novel architecture designed to address three video moment retrieval tasks (NLMR, SLMR and MR+HD).
The key innovation lies in integrating a retention mechanism into the multimodal Transformer framework, resulting in the Maskable Retention Transformer (MRT) block.
The MRT block divides the language and video attention map into four regions $\mathcal{A}(q \rightarrow q)$, $\mathcal{A}(q \rightarrow v)$, $\mathcal{A}(v \rightarrow q)$ and $\mathcal{A}(v \rightarrow v)$.
The primary contributions of this work are the introduction of the bidirectional retention mask, represented by matrix $\mathcal{M}{RD}$,  and the learnable sparse retention mask, denoted by matrix $\mathcal{M}_{LS}$, which are specifically designed for enhanced video modeling.
Experimental results on five benchmark datasets, particularly ActivityNet Captions, demonstrate the effectiveness of this method.

**Strengths:**

- The main innovation of the article lies in the bidirectional retention mask and the learnable sparse retention mask. The design ideas are very intuitive, aiming to model the bidirectional, decaying, and nonlinear temporal associations between video sequences, as well as to adaptively learn strong associative responses related to the target event and eliminate redundant background interference in the video.
- The method is simple, yet the performance improvement is obvious.  Meanwhile, this method achieves a trade-off between model size and accuracy.
- From the description, the structure does not seem complicated and should be easy to reproduce. However, it would be great if the author could open source the implementation in the future.
- The author provides supplementary materials that discuss more experimental results.

**Limitations:**

- Despite the intuitive motivation of the learnable sparse retention mask, the implementation details are not convincing. Why does Eq. 5 achieve the effect of reducing redundant noise interference and learning more critical information under the supervision of Eq. 6? I have not found any explanation in the article or supplementary materials about how the sparse ratio $\alpha$ controls the proportion of responses.
- The main innovation of the article is the introduction of the bidirectional retention mask and learnable sparse retention mask. However, the baseline in Table 6 on the ANetCap dataset has already achieved state-of-the-art performance. What has led to such strong performance?
- The ablation experiment R2.1 only compares the performance difference between bidirectional and unidirectional, but it does not explain how unidirectional is implemented, such as how to adjust Eq.4 to make it unidirectional?
- There is an issue with Eq.4. $(\mathcal{M}_{RD})_{n,m}$ should be a scalar, but $\mathcal{M}_{RD} \in \mathbb{R}^{T \times T}$.
- The sparsity constraint loss in Table 8 should be $\mathcal{L}_{sparse}$.
- Here are some suggestions: ① It would be helpful to annotate that Figure 2 displays the experimental results of the MRNet in multi-query training mode on the ANetCap dataset, to prevent any confusion; the methods and experimental analysis sections could be expanded with more detail, as the introduction is too verbose - focus on the key points; the right figure of Figure 7 could benefit from a more detailed explanation.

**Suitability:**

3

---

### Meta-Review · Area_Chair_eASG · 2024-07-01

**Recommendation:** Accept (Poster)
**Confidence:** 5

**Metareview:**

In summary, the reviewers highlight several strengths of the paper, including its innovative bidirectional retention mask and learnable sparse retention mask, intuitive design for modeling temporal associations in video sequences, significant performance improvement, and comprehensive experimental validation. Minor concerns are raised regarding the clarity and implementation details of the sparse retention mask, the complexity and readability of the paper, the lack of open-sourced code for reproducibility, and specific issues with formula definitions, formatting, and comprehensive explanations of certain methodological choices.

Overall, two reviewers lean towards acceptance, while one leans towards rejection due to presentation issues. Considering this, I recommend accepting the manuscript for publication. However, please revise your manuscript further based on the reviewer's comments on the writing.